# Recording of Cardiac Excitation Using a Novel Magnetocardiography System with Magnetoresistive Sensors Outside a Magnetic Shielded Room

**DOI:** 10.3390/s25154642

**Published:** 2025-07-26

**Authors:** Leo Yaga, Miki Amemiya, Yu Natsume, Tomohiko Shibuya, Tetsuo Sasano

**Affiliations:** 1Department of Cardiovascular Medicine, Institute of Science Tokyo, Tokyo 113-8519, Japan; yaga.leo@tmd.ac.jp (L.Y.); amemiya.miki@tmd.ac.jp (M.A.); 2Advanced Products Development Center, Technology and Intellectual Property HQ, TDK Corporation, Tokyo 103-6128, Japan; yu.natsume@tdk.com (Y.N.); tomohiko.shibuya@tdk.com (T.S.)

**Keywords:** magnetocardiography, signal processing, noise reduction, medical sensing

## Abstract

**Highlights:**

**What are the main findings?**
A novel MCG system using room-temperature magnetoresistive sensors was developed.The system reliably captured cardiac signals without magnetic shielding.

**What is the implication of the main finding?**
Enables cost-effective and portable MCG suitable for clinical environments.Opens new possibilities for noninvasive cardiac diagnostics and monitoring.

**Abstract:**

Magnetocardiography (MCG) provides a non-invasive, contactless technique for evaluating the magnetic fields generated by cardiac electrical activity, offering unique spatial insights into cardiac electrophysiology. However, conventional MCG systems depend on superconducting quantum interference devices that require cryogenic cooling and magnetic shielded environments, posing considerable impediments to widespread clinical adoption. In this study, we present a novel MCG system utilizing a high-sensitivity, wide-dynamic-range magnetoresistive sensor array operating at room temperature. To mitigate environmental interference, identical sensors were deployed as reference channels, enabling adaptive noise cancellation (ANC) without the need for traditional magnetic shielding. MCG recordings were obtained from 40 healthy participants, with signals processed using ANC, R-peak-synchronized averaging, and Bayesian spatial signal separation. This approach enabled the reliable detection of key cardiac components, including P, QRS, and T waves, from the unshielded MCG recordings. Our findings underscore the feasibility of a cost-effective, portable MCG system suitable for clinical settings, presenting new opportunities for noninvasive cardiac diagnostics and monitoring.

## 1. Introduction

Magnetocardiography (MCG) is a technique employed to detect the magnetic fields generated by electrical currents in the heart. Previous research has highlighted several advantages of MCG compared to electrocardiography (ECG) [1,2]. ECG records the electrical activities of the heart by measuring the electrical potential difference between electrodes attached to the skin. Consequently, the signals are influenced by the heterogeneous conductivity of body tissues and the resistance at the electrode interface [3,4]. In contrast, magnetic permeability is relatively uniform across muscles, bones, skin, and fluids in the human body and in the air. Therefore, MCG can capture signals from all directions, including those from the back, whereas such signals are attenuated in ECG [5,6,7,8]. In MCG, sensors can be aligned in an array to perform multipoint recordings. The integration of these features contributes to the analysis of the excitation of the heart through three-dimensional identification of electrical activities. Furthermore, measurement of the magnetic field does not require direct contact between sensors and the body, thereby eliminating the risk of skin irritation or allergic reactions [9]. Given these attributes, MCG has been utilized to record fetal cardiac excitation. Capturing the electrical activities of the fetal heart is challenging due to the insulating effects of the fetal vernix and maternal tissues [10,11]. Therefore, MCG is a promising technique for assessing fetal cardiac activity.

Magnetic fields generated by cardiac electric currents are extremely weak (50–100 pT) compared with geomagnetic fields (25–65 µT) and other environmental magnetic fields [12,13]. The most prevalent device for MCG recording is the superconducting quantum interference device (SQUID) sensor, which consists of a superconducting loop with two Josephson junctions, whereby the outer magnetic flux induces an oscillating voltage that is converted into output signals [14]. Due to the high sensitivity and measurement range of the SQUID sensor, a magnetic shielded room (MSR) is required to prevent contamination by environmental magnetic noise. Additionally, the SQUID sensor requires cooling using liquid helium. These requirements pose significant challenges to the widespread clinical application of MCG systems.

The magnetoresistive (MR) effect is a property of materials that alters their electrical resistance in response to an external magnetic field [15,16]. We developed novel magnetoresistive sensors with high sensitivity and a large dynamic range that can be operated at ambient temperatures, without necessitating cooling systems.

In this study, we aimed to utilize novel MR sensors to record the magnetocardiograms in a normal environment without an MSR. We combined signal averaging and signal processing methods to mitigate the substantial environmental noise resulting from the absence of an MSR. The objective of this study was to obtain an MCG signal comparable to an ECG signal outside of MSR.

## 2. Materials and Methods

### 2.1. Participants

Magnetocardiograms of 40 healthy participants were recorded in an examination room at the Institute of Science Tokyo Hospital. Patients with sustained arrhythmia or other cardiac diseases were excluded from the study. The study was approved by the Institutional Review Board of the Institute of Science Tokyo (M2023-037).

### 2.2. Experimental System Design

A newly invented MR sensor with high sensitivity and wide dynamic range (Nivio xMR sensor, TDK Corporation, Tokyo, Japan) was used in this study. The MR sensor element consists of thin layers of ferromagnetic and nonferromagnetic conductive materials [17]. Compared to conventional MR sensors, this sensor offers the following novel features:Ultra-high sensitivity and low noise characteristics enable highly accurate detection of weak magnetic signals.The design is optimized for biomagnetic applications, making it possible to detect biomagnetic signals and magnetic nanoparticles.Sensor array integration allows for simultaneous multi-point detection and high spatial resolution.The wide dynamic range and excellent linearity enable quantitative measurements without the need for magnetic shielding.

The electrical resistance of the sensor element varies depending on the angle between the sensor axis and the projection of the external magnetic fields, converting changes in the magnetic field into electrical signals. Four sensor elements were assembled to form a Wheatstone bridge to improve the signal-to-noise ratio [18]. This sensor was rectangular-cuboid in shape (12 × 12 × 74 mm) and detected magnetic fields along the long axis of the sensor. A magnetic circuit inside the sensor case enables preferential detection of the magnetic field component along the longitudinal axis. The noise density of the MR sensor measured in an MSR was 3.1 pT/√Hz at 1 Hz at optimal performance, with a dynamic range from −45 µT to 45 µT.

The measurement system developed for this study was termed the seat-type outside-MSR MCG system (STORM system) (Figure 1). In the STORM system, the MR sensors were arrayed on the back of the seat. The sensor module and seat were separated to prevent contamination by vibration noise from participants. The sensor array comprised 42 MR sensors (configured in a 6 × 7 matrix with a 4.0 cm interval between adjacent sensors) and was positioned 5 mm from the surface of the seat. In addition to the 42 MR sensors for recording, the STORM system was equipped with 21 reference sensors surrounding the recording sensor module to detect environmental magnetic fields. An ECG module was introduced to record a 1-channel ECG. The signals from the STORM system were recorded and analyzed using the software developed by TDK Corp (Tokyo, Japan).

### 2.3. Study Protocol

MCG and ECG recordings were obtained using the STORM system. The sampling rate was set at 5000 Hz, with a recording duration of 120 s. This sampling rate allows for more accurate capture of fine waveform details, including rapid rises and falls in the signal. In addition, acquiring data at a high sampling rate increases the flexibility of post-processing, such as the application of digital filters and noise reduction, and helps to prevent aliasing. Empty room noise and noise data of the environmental magnetic fields were recorded once daily. After removing any magnetic objects, the participants sat down facing the rear of the seat.

Non-magnetic electrodes (V-09IO3, NIHON KODEN, Tokyo, Japan) with conductive adhesive gel and carbon cables were used for ECG measurements. The ECG circuit was a custom-built two-lead electrocardiograph, designed with reference to Texas Instruments ICs and application notes (https://www.ti.com/solution/en-jp/electrocardiogram-ecg#tech-docs, accessed on 14 June 2025). Analog signals were digitized using an NI-9202 A/D converter (National Instruments, Austin, TX, USA). Subjects were recorded in a resting state, and the electrodes were placed in a standard two-lead configuration. The ECG was recorded simultaneously with the MCG measurement for 2 min.

The obtained ECG data were used for comparison with the MCG data. MCG and ECG signals were acquired using a custom C++ program that calls the National Instruments (NI) device driver and were saved as data files.

### 2.4. Signal Processing

After recording, several signal processing techniques were employed to extract the MCG waveforms (Figure 2). Initially, the raw signals were filtered using a bandpass filter (0.3–40 Hz), followed by a notch filter (50 Hz). Adaptive noise cancelling (ANC) was then performed to reduce the influence of environmental magnetic field components on the MCG channels, utilizing readings from reference sensors. A detection magnetic sensor acquires the primary signal, which contains both the desired signal and noise, while a reference magnetic sensor acquires a signal that primarily contains noise similar to that affecting the detection sensor. Using an adaptive algorithm, the noise component from the reference sensor is estimated and subtracted from the primary signal, thereby effectively reducing noise while preserving the desired signal. This approach enables more accurate extraction of the target signal from a noisy environment [19]. The environmental signals were projected onto the MCG sensor subspace using an adaptive filter and subsequently subtracted [20].

After noise cancellation using ANC, we performed further signal-processing procedures consisting of signal averaging and noise reduction. First, the peaks of the R waves were detected in simultaneously recorded ECG, and MCG data after ANC were signal-averaged based on the detected R wave timings. Subsequently, a novel noise reduction method termed Bayesian signal space projection (SSP) was applied to the averaged MCG waveforms. The empty room noise data were used to identify the spatial characteristics of the noise to be discriminated from the MCG data. This algorithm was based on a model expressed using the following formula:$$y(t)\;=\;y_{S}(t)\;+\;y_{I}(t)\;+\;\varepsilon$$
where *y(t)* is a vector representing measured values of the MCG sensor array at time t. *y_S_(t)* is the signal vector containing the spatial information of the signal of interest. *Y_I_(t)* represents the interference vector generated from sources outside the signal sources. ε denotes stationary noise that is uncorrelated across channels [21,22]. Both *y_S_(t)* and *Y_I_(t)* are components projected from factors, defined as *y_S_(t)* = A *x(t)*, where *A* is the signal factor loading matrix and x(t) is the signal factor, and *Y_I_(t)* = B *u(t)*, where B is the interference factor loading matrix and u(t) is the interference factor. In PFA [cite-pfa], B is determined by applying the factor analysis method to ERN data. Once B is determined, A is estimated by applying the factor analysis method to MCG data, which includes both MCG and interference, by modeling the total factor loading matrix as [A; B], with B fixed to the value determined from the ERN analysis. With A and B determined, we can estimate *x(t)* and consequently the denoised signal component, *y_S_(t)*. The recorded data files were subsequently imported into a Python-based analysis environment (version 3.10.13). Data analysis was performed by executing a series of custom Python scripts, with processing parameters pre-defined by the user and uniformly applied to all datasets. To ensure consistency and reproducibility of the analysis across all subjects, batch processing was employed, and the generation of plots and the calculation of metrics such as signal-to-noise ratio (SNR) were automated.

### 2.5. Data Analysis

To evaluate the quality of the MCG signal, the signal-to-noise ratio (SNR) was calculated using the formula below. The baseline was manually determined using simultaneously recorded ECG and defined as the 100 ms period before the onset of the P wave [23].$$SNR=20{log}_{10}\frac{QRS\;amplitude}{baseline\;amplitude}$$

Isomagnetic field maps were constructed by arranging the signal data at each point in time according to the positions of the sensor array and complementing values between sensors.

### 2.6. Statistical Analysis

Statistical analyses were performed using the EZR software (version 1.68, Saitama Medical Center, Jichi Medical University, Saitama, Japan), which is a graphical user interface for R (version 4.3.0, The R Foundation for Statistical Computing, Vienna, Austria). Data are expressed as mean ± standard deviation (SD). Student’s *t*-test or Welch’s *t*-test was used for between-group comparisons. Statistical significance was set at *p* < 0.05.

## 3. Results

### 3.1. Noise Reduction by ANC

Forty participants were enrolled in this study. The participant characteristics are presented in Table 1.

Initially, we attempted to reduce the environmental noise by applying ANC, a newly invented noise-cancelling method. The noise spectral density of the representative channel exhibited a broad range of noise reduction using ANC (Figure 3a). The noise density at 1 Hz decreased from 344.8 pT√Hz to 15.0 pT√Hz. This noise-reduction procedure could remove large amounts of noise emitted from the surrounding magnetic materials and geomagnetic fields. The representative waveform of the magnetic signal exhibited an alternation of the waveform caused by environmental noise, which was eliminated by the application of ANC (Figure 3b).

### 3.2. Noise Reduction by Signal-Average and Bayesian SSP

The signal-averaged MCG waveform from a single representative channel through each stage of signal processing illustrates a progressive reduction in noise (Figure 4a–c). After the application of the digital filter alone, the raw waveform exhibited considerable noise contamination (Figure 4a). The application of ANC substantially reduced the background noise, enabling the visualization of the QRS complex and T waves; however, P waves with small amplitudes were unclear (Figure 4b). The application of Bayesian SSP further eliminated the residual noise component and flattened the baseline of the waves, enabling detection of the P wave (Figure 4c). Compared to a simultaneously recorded ECG, the application of our noise reduction methods made the MCG waveform of comparable quality as the QRS complex. However, the P and T amplitudes were smaller in the MCG than in the ECG (Figure 4c,g). The waveforms of all 42 channels overlapped (Figure 4d–f), indicating that noise reduction was achieved in all channels.

We recorded the MCG during sinus rhythm using these noise reduction methods in 40 participants and calculated the SNR for the QRS complex in each case (Table 2). Noise reduction was applicable to all participants regardless of individual differences. The noise reduction procedures significantly improved the SNR from the raw waveform with digital filter, application of ANC, and further application of Bayesian SSP (9.8 ± 3.7, 25.9 ± 4.8, and 35.0 ± 5.2 dB, respectively). The SNR after the application of Bayesian SSP was equivalent to that of the ECG (37.9 ± 6.5 dB).

Using novel noise-reduction methods, we recorded the MCG without an MSR and displayed the waveform according to the sensor array (Figure 5a). The amplitude and direction of the MCG signals were different in the individual channels because of the distance and direction between the sensor and the current source. The waveforms of representative channels (ch.2 and ch.21) are shown in Figure 5b,c. The directions of the QRS complex and T wave were opposite; however, both waveforms showed visible P, QRS complex, and T waves.

### 3.3. Isomagnetic Field Maps During Signal Processing

A distinctive feature of MCG is the use of multipoint mapping to construct isomagnetic field maps, which enable the evaluation of cardiac excitation, including its spatial distribution. However, the contamination of external noise causes large errors in the calculation of the isomagnetic field map. In this study, we constructed an isomagnetic field map using MCG waveforms with digital filters only, application of ANC and signal averaging, and application of Bayesian SSP (Figure 6). Noise reduction with ANC improved the homogeneity of the map at baseline, which contributed to the evaluation of the magnetocardiograms with spatial information. However, further smoothing of the waveforms by Bayesian SSP did not appear to significantly affect the map pattern, focusing on the QRS complex.

## 4. Discussion

This study utilized a newly developed MR sensor and novel signal processing techniques to reduce environmental noise. Reference sensors were employed, leveraging the wide dynamic range of the MR sensor, and ANC was applied to eliminate environmental noise, including that of the geomagnetic field. The integration of ANC, signal averaging, and Bayesian SSP facilitated the acquisition of MCG waveforms without the need for an MSR, achieving a high SNR comparable to that of the conventional ECG. Each component of the waveform, particularly the T wave, was sufficiently smoothed to calculate the parameters for evaluation following Bayesian SSP.

MCG has been under development for several years. A magnetic field can penetrate various substances, such as water, air, or space, without interference, and can be detected without direct contact. In addition, MCG can obtain spatial information of the signal source by arranging multiple sensors as an array. Given that MCG has the potential to provide more information on cardiac excitation than ECG, its clinical application has been anticipated [24]. SQUID sensors have primarily been used due to their high sensitivity and consequent ability to detect small magnetic fields resulting from cardiac excitation.

Despite the potential of MCG, its clinical applications have been constrained. The main challenge has been the difficulty in detecting the small signals generated by cardiac excitation in the presence of substantial environmental noise.

In typical settings, magnetic fields from surrounding materials are also detected by the sensor, obscuring cardiac signals. Notably, the Earth’s magnetic field is considerably stronger than that of the heart [25]. Although the SQUIDs possess the required sensitivity to detect cardiac magnetic fields, they must be employed in an MSR because the environmental magnetic fields exceed the dynamic range of the sensor. Furthermore, the sensor requires a cryogenic cooling system to maintain superconductivity, only operating at cryogenic temperatures achieved with liquid helium. Because of these constraints, the installation of an MCG system demands substantial equipment, an MSR, and a cooling system. There is a report of MCG measured by SQUIDs, which work at relatively high temperature at 77 K with liquid nitrogen [26,27]. Nevertheless, the sensor noise of high-Tc SQUIDs at the temperature interferes with recording in unshielded environments [28].

In addition to the SQUID, several sensors have been developed for MCG recording. Spin-exchange-relaxation-free (SERF) optically pumped magnetometer (OPM) has a compact sensor size and does not require cooling systems [29]. However, the SERF-OPM possesses a narrow dynamic range and requires magnetic shielding to remove the surrounding magnetic field, similar to a SQUID, and is known to exhibit a limited frequency bandwidth [30,31]. Several studies utilizing induction coils, fluxgate sensors, and scalar OPMs have reported that they do not require magnetic shielding or cooling systems [4,32,33,34]. However, the performance of these sensors is limited in unshielded environments, complicating the recording of cardiac magnetic field signals. In this study, we used a novel MR sensor. Although a previous study used an MR sensor array to measure the cardiac magnetic field, the measurements were conducted inside an MSR [35]. The novel MR sensor used in the present study possessed a wide dynamic range, enabling simultaneous recording of cardiac excitation and environmental noise. Therefore, by integrating multiple signal processing techniques to remove the environmental noise, high-quality MCG signals were extracted without utilizing an MSR.

In this study, ANC-employing reference sensors were implemented to eliminate substantial environmental noise. This technique has been extensively applied to remove environmental noise from microphone voice signals [36] and eliminate artifacts caused by patient movements from ECG signals [37]. ANC is known to be effective in situations where the noise is emitted from materials outside the participant’s body and is unrelated to the magnetic signal from the heart [38]. The performance of ANC depends on the positioning of the target and reference sensors. Additionally, because MR sensors detect uniaxial changes in the magnetic field, the orientation of the reference sensor affects the effectiveness of noise cancellation. In our study, the number and position of the reference sensors were optimized to suppress noise levels.

In the MCG waveforms obtained through the application of ANC and averaging, the QRS complexes were distinctly identifiable. However, residual noise persisted in the baseline and T-wave segments, and the P waves were unclear. Consequently, Bayesian SSP was applied to further eliminate noise. Bayesian SSP is a signal processing technique that differentiates between signals and noise based on differences in their spatial characteristics. Various magnetic objects surrounded the sensor, transmitting noise from all directions, whereas the cardiac activity signal originated from a specific location. In the Bayesian SSP procedure, the spatial information of the environmental noise obtained by empty room noise in the absence of the subject was utilized to subtract the noise from the MCG recording. Because the empty room noise was measured once daily, prior to the experiment, its spatial characteristics did not necessarily entirely correspond with the noise at the actual time of measurement. However, the baseline noise was reduced, and the P wave became discernible after applying Bayesian SSP. The SNR for the QRS complex was further improved, reaching a level comparable to that of the ECG. In this study, the amplitudes of the P and T waves in the magnetocardiogram (MCG) were lower than those in the electrocardiogram (ECG). This discrepancy can be attributed to two main factors. First, the ECG recordings were obtained using lead II, a lead commonly used for its clear visualization of P and T waves, whereas the MCG did not necessarily utilize a channel optimized for the visibility of these waves. Second, in the MCG analysis, the channel was selected based on the maximum QRS amplitude, without specific consideration for the clarity or magnitude of the P and T waves.

Regarding the clinical utility of MCG, studies have indicated that MCG can be used to diagnose ischemic heart disease by evaluating its waveform and isomagnetic field map [39,40]. Additionally, several studies have explored methods for localizing abnormal electrical activity in the heart by estimating current sources through multipoint mapping, integrated with three-dimensional anatomical information obtained from magnetic resonance imaging or X-ray computed tomography [41,42,43]. It remains uncertain whether the signal accuracy of MCG obtained without MSR in this study is sufficient for these technologies; however, future applications are anticipated.

This study possesses some limitations. First, R-wave-triggered averaging was performed, thereby limiting MCG recordings to sinus rhythms and precluding the detection of arrhythmias. Moreover, although we have confirmed that it is possible to measure magnetic fields when there is a change in external noise, such as a person walking around the examination room, it is impossible to extract magnetic field signals when sudden noise is introduced because external noise is canceled using the empty room noise obtained in advance. In addition, the position of the sensor array was fixed in the STORM system, and the results varied depending on the physique and posture of individuals. The quality of the results could be enhanced by further improvement of the seat that allows for positional adjustments of the sensor array or the participant during MCG recording. Furthermore, the majority of the participants are male, and elderly subjects have not been included in this research. However, though the mean SNR of MCG signals of female participants (30.1 dB) was relatively lower than that of men (35.8 dB), the analysis could be performed without problems for women as well. In addition, we are including even older subjects in our current investigation, and data are recorded without any problems. We would like to clarify this in the future. Additionally, the sample size was relatively small (n = 40), necessitating further studies with larger cohorts, especially for analysis with deep learning [44].

## 5. Conclusions

The integration of a novel MR sensor possessing a wide dynamic range and the application of optimized noise reduction techniques enabled the successful recording of MCG signals without the need for an MSR. Furthermore, by applying signal averaging, a signal quality (SNR: 35.0 ± 5.2 dB) comparable to that of an ECG (SNR: 37.9 ± 6.5 dB) was achieved. These findings demonstrate the feasibility of this approach to facilitate extensive clinical applications of MCG.

## Figures and Tables

**Figure 1 sensors-25-04642-f001:**
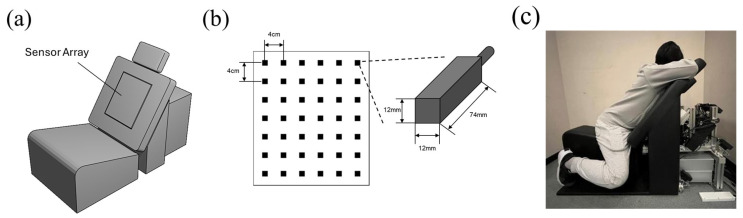
Magnetoresistive sensors and the MCG system. (**a**) Illustration of the STORM system; (**b**) configuration of the sensor array; and (**c**) appearance during measurement.

**Figure 2 sensors-25-04642-f002:**
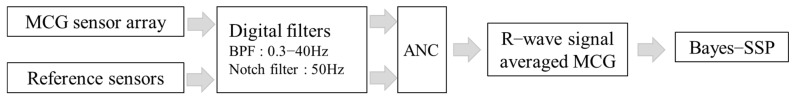
A diagram illustrating procedures for signal processing.

**Figure 3 sensors-25-04642-f003:**
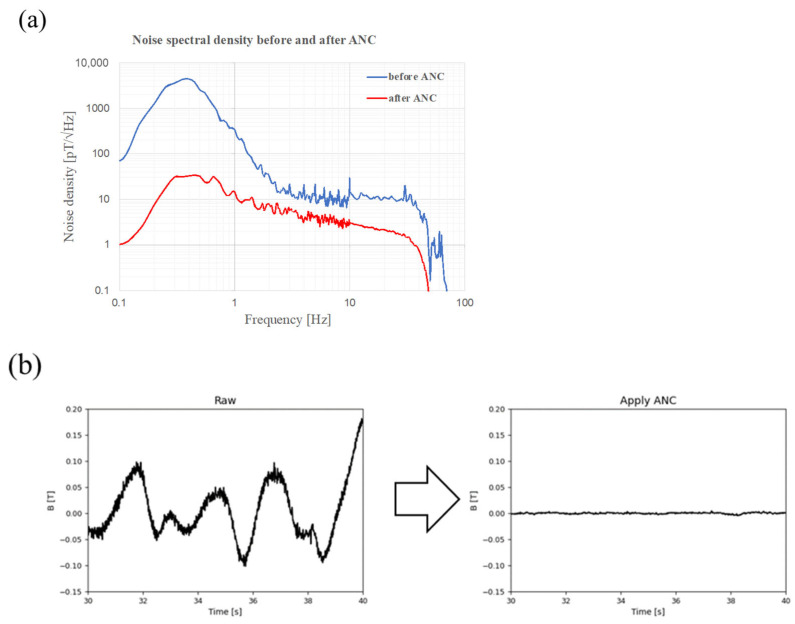
Noise reduction effect by adaptive noise cancellation (ANC). (**a**) Noise spectral density before (blue line) and after (red line) application of ANC. The application of ANC reduced the environmental noise level throughout the whole frequency range. (**b**) The waveform measures environmental signals from one representative channel. The waveform before applying ANC shows alternation in signals (left). The application of ANC eliminates most signals, resulting in the flat waveform (right).

**Figure 4 sensors-25-04642-f004:**
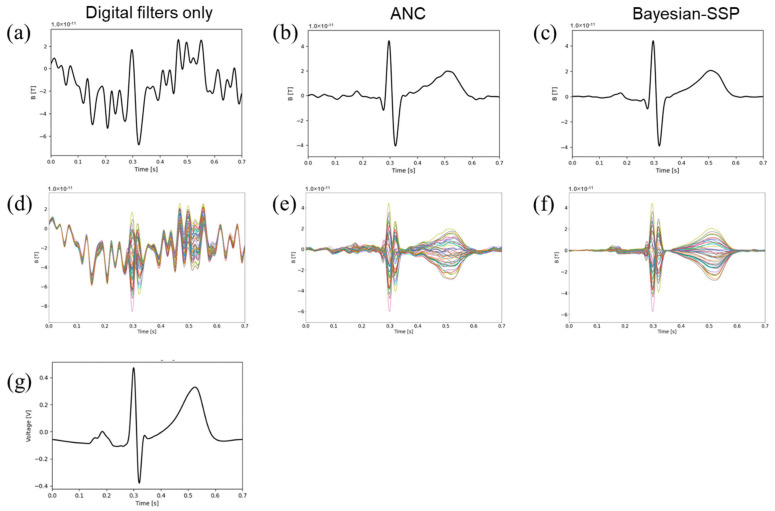
Effect of noise reduction in averaged MCG waveforms after ANC. Waveform of MCG in representative 1 channel (ch.9 of ST-3) is shown after digital filters only (**a**), after application of ANC and signal averaging (**b**), and further application of Bayesian SSP (**c**). (**d**–**f**) The overlapping waveforms of all 42 channels are shown in (**a**–**c**). The waveforms from each channel are drawn in different colors. (**g**) Waveform of lead II ECG recorded simultaneously with the MCG.

**Figure 5 sensors-25-04642-f005:**
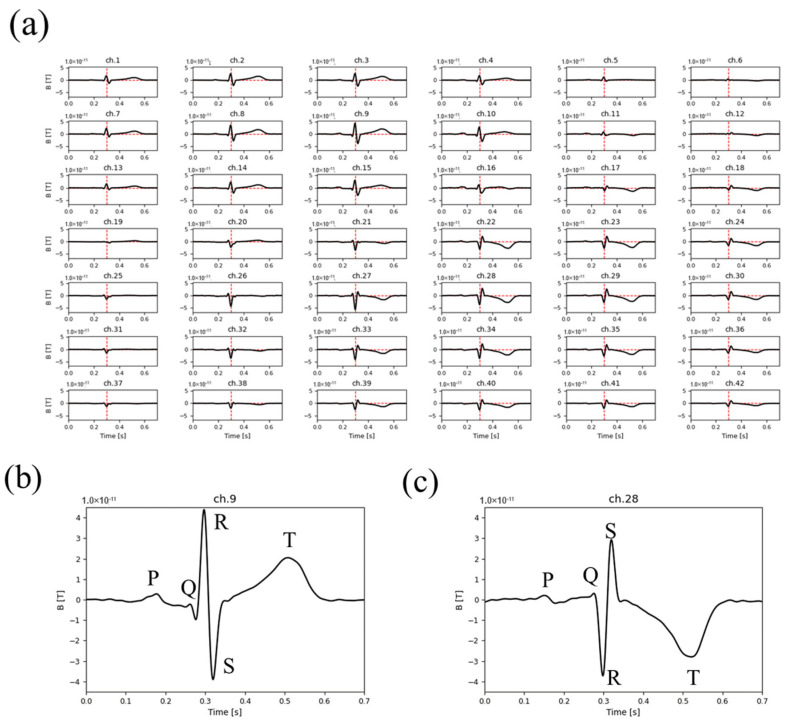
Waveform array of MCG after the noise reduction process. (**a**) Waveforms of each magnetoresistive sensor are displayed in the configuration of the sensor array. (**b**,**c**) Enlarged waveforms of representative channels selected from the waveform array. Waveforms of Ch. 9 (**b**) and Ch. 28 (**c**) are exhibited. The component of the wave (P, Q, R, S, and T) is indicated in each figure.

**Figure 6 sensors-25-04642-f006:**
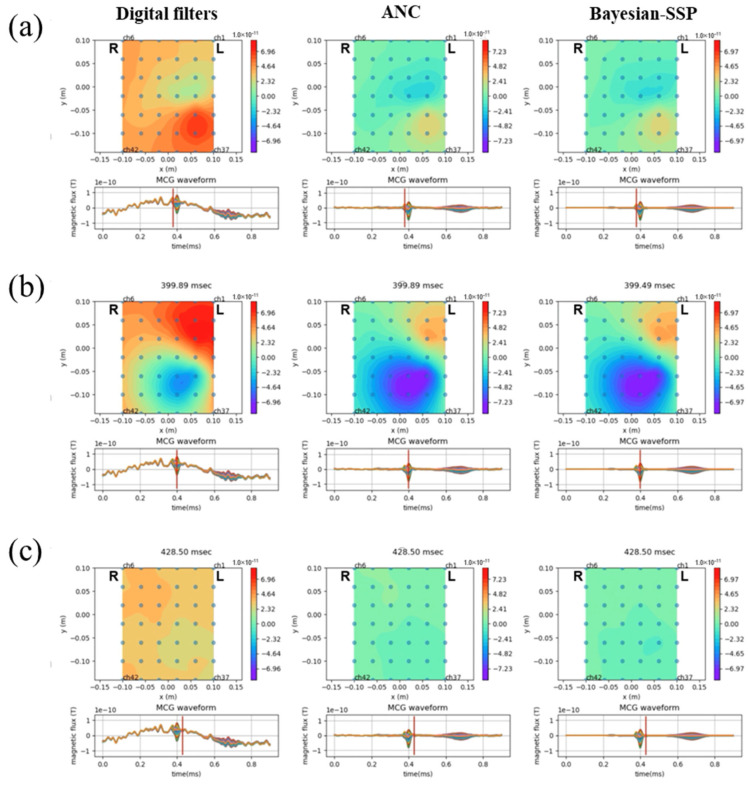
Isomagnetic field maps after noise reduction processes. Isomagnetic field maps are shown at the timing of the Q-waves (**a**), R-waves (**b**), and the end of the QRS complex (**c**). Each image was constructed using MCG waveforms after digital filters (left), application of ANC and signal averaging (center), and further application of Bayesian SSP (right), respectively.

**Table 1 sensors-25-04642-t001:** Characteristics of participants.

	Participants (*n* = 40)
Age, years	44.1 ± 10.5
Sex, n	
Male	34
Female	6
Height, cm	169.3 ± 7.4
Weight, kg	62.1 ± 9.2
BMI, kg/m^2^	21.6 ± 2.7

**Table 2 sensors-25-04642-t002:** Signal-to-noise ratio of each participant after signal processing.

Participant No.	Digital Filters (dB)	ANC (dB)	Bayesian SSP (dB)	ECG (dB)
#01	16.2	31.0	34.4	35.3
#02	13.1	28.8	37.0	43.8
#03	12.5	28.5	43.3	40.3
#04	8.2	26.6	38.9	53.3
#05	13.1	20.8	29.9	28.5
#06	13.3	25.8	34.6	32.4
#07	16.4	34.1	41.3	39.3
#08	9.9	22.8	36.1	40.2
#09	12.9	23.5	28.5	32.7
#10	12.1	26.2	32.9	26.5
#11	5.7	25.9	42.7	54.0
#12	11.1	22.5	35.1	28.6
#13	13.5	25.4	36.2	37.9
#14	12.6	26.6	30.2	35.1
#15	10.0	26.9	28.5	31.8
#16	12.4	27.3	44.3	41.7
#17	3.6	21.6	23.3	32.9
#18	6.0	26.6	35.4	42.5
#19	7.5	28.1	37.6	42.0
#20	2.4	18.7	30.9	29.1
#21	13.3	19.4	31.2	35.8
#22	5.7	20.5	27.2	46.3
#23	12.2	26.7	36.8	40.7
#24	5.7	18.3	28.4	40.8
#25	13.8	32.8	36.1	34.3
#26	10.9	31.0	39.0	31.5
#27	7.0	27.1	38.3	42.9
#28	11.8	33.5	35.5	36.6
#29	4.3	29.8	41.8	31.6
#30	8.1	24.6	31.2	37.6
#31	9.5	20.4	28.1	37.9
#32	7.2	23.7	32.1	26.5
#33	2.8	27.1	38.8	41.2
#34	6.6	27.5	37.6	46.6
#35	4.2	11.1	22.0	34.2
#36	11.4	28.0	41.4	39.4
#37	12.0	22.4	36.8	43.0
#38	13.2	32.1	37.7	38.3
#39	13.1	29.8	38.4	48.0
#40	7.0	33.7	36.8	35.7
Average	9.8 ± 3.7	25.9 ± 4.8	35.0 ± 5.2	37.9 ± 6.5

## Data Availability

The original contributions presented in this study are included in the article. Further inquiries can be directed to the corresponding author(s).

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
