# Peer review of "Recording of Cardiac Excitation Using a Novel Magnetocardiography System with Magnetoresistive Sensors Outside a Magnetic Shielded Room"

_sensors, 2025, doi:10.3390/s25154642_

Round 1
Reviewer 1 Report
Comments and Suggestions for Authors
My comments describe in the Referat.

Author Response
Response to Reviewer 1
The article is devoted to the development of a new magnetocardiographic system with magnetoresistive sensors, which allows recording cardiac signals without magnetic shielding. The SQUID sensors used to obtain MCG are known to require magnetic shielding and liquid helium. The authors developed new magnetoresistive sensors with a sensitivity of 3.1 pT/√Hz and, based on them, presented the STORM system, which made it possible to obtain MCG of patients without using magnetic shielding. It should be noted that an effective noise suppression system was used. Comparison of the quality of the obtained MCG signals with ECG signals showed the high efficiency of the proposed method and that it can have wide clinical application.
At the same time, the reviewer has several questions.
- How did the developed sensor measure magnetic fields oriented only along the long axis of the sensor?
This sensor is designed with a magnetic circuit inside the case that enables preferential detection of the magnetic field component along the longitudinal axis. While the detailed internal structure cannot be disclosed due to proprietary company know-how, this magnetic circuit design effectively suppresses the influence of magnetic field components from other directions and allows for highly sensitive measurement of the longitudinal magnetic field component. We added this information in Section 2.2.
- The authors need to explain in more detail the operation of the adaptive noise suppression system.
In this system, adaptive noise cancellation (ANC) is implemented using two types of sensors: a detection sensor and a reference sensor. The detection sensor measures both the weak magnetic signals originating from the heart and environmental noise, while the reference sensor primarily detects only the environmental noise. The ANC algorithm uses the environmental noise signal obtained from the reference sensor to estimate and remove the environmental noise component in the detection sensor signal in real time. As a result, it is possible to extract only the cardiac magnetic signal with high accuracy. We added the explanation in Section 2.4.
- The authors need to explain the operation of the software product that recorded and analyzed signals from the STORM system.
Thank you for your valuable comments. According to your comments, we have added the following descriptions to Section 2.3 and Section 2.4:
[Addition to Section 2.3. Study Protocol]
MCG and ECG signals were acquired using a custom C++ program that calls the National Instruments (NI) device driver, and were saved as data files.
[Addition to Section 2.4. Signal Processing]
The recorded data files were subsequently imported into a Python-based analysis environment. Data analysis was performed by executing a series of custom Python scripts, with processing parameters pre-defined by the user and uniformly applied to all datasets. To ensure consistency and reproducibility of the analysis across all subjects, batch processing was employed, and the generation of plots as well as the calculation of metrics such as signal-to-noise ratio (SNR) were automated.
Reviewer 2 Report
Comments and Suggestions for Authors
Submitted manuscript is devoted to the development of the magnetoresistive sensor array operating at room temperature. The subject is important and selection of the journal is appropriate. However, despite the general evaluation of the work as high-quality research, some mandatory changes must be made in order to improve quality of the manuscript and bliss it toward the overall quality of publications in this journal.
- The electromagnetic activity of the heart was known long ago and some general references related to biophysics (Roland Glaser or similar) would be an advantage.
- Authors make their focus on the fact that they use newly invented MR sensor (https://doi.org/10.3390/chemosensors9080211) in this case they must explain clearly the degree of the novelty obtained in the present studies.
- Work has no original graphical data except graphs. The photograph of the device and schematic description of the measurements involving patients would be an advantage.
- Utilization of the magnetic field sensors of different types for detection and evaluation of caddio-vascular signals was previously widely discussed in the literature. Comparative general analysis and discussion around this problem might be important for this journal audience as different kind of magnetic sensors have both particular advantages and disadvantages starting from magnetic field sensitivity (Uchiyama, T.; Mohri, K.; Honkura, Y.; Panina, L.V. Recent advances of pico-Tesla resolution magneto-impedance sensor based on amorphous wire CMOS IC MI Sensor. IEEE Trans. Magn. 2012, 48, 3833–3839; Prinz, G.A. Magnetoelectronics applications. Magn. Magn. Mater.1999, 200, 57–68; Kozlov, N.V., Volchkov, S.O., Blyakhman, F.A., Chestukhin, V.V., Kurlyandskaya, G.V. The Modeling of Magnetic Detection of Iron Oxide Nanoparticles in the Stream of Patient-Specific Artery With Stenotic Lesion: The Effects of Vessel Geometry and Particle Concentration IEEE Transactions on Magnetics, 2022, 58(8), 5100305). There were excellent MCG measurements with portable SQUIDs like - Impedance magnetocardiography: Experiments and modeling by Claycomb. Discussion using this experience might be just mor convincing.
- From the description of the patients, it becomes not very clear if they are supposedly healthy or having identified cardiac problems. In the last case the averaging should be done separately.
- Figures 5 and 6 have clear technical deficiencies – they are not readable. Fig.5 part a) should be the double size; parts b) and c) maybe placed together in the next row. Fig.6 should be the double size of the present dimension.
- Conclusions must contain numerical data.
Author Response
Reviewer 2
Submitted manuscript is devoted to the development of the magnetoresistive sensor array operating at room temperature. The subject is important and selection of the journal is appropriate. However, despite the general evaluation of the work as high-quality research, some mandatory changes must be made in order to improve quality of the manuscript and bliss it toward the overall quality of publications in this journal.
1. The electromagnetic activity of the heart was known long ago and some general references related to biophysics (Roland Glaser or similar) would be an advantage.
Thank you for your valuable comments. We added the reference according to your suggestion (Ref. 1).
2. Authors make their focus on the fact that they use newly invented MR sensor (https://doi.org/10.3390/chemosensors9080211) in this case they must explain clearly the degree of the novelty obtained in the present studies.
Thank you for your valuable comments. The MR sensor used in this study is the Nivio™ XMR sensor developed by TDK (https://product.tdk.com/en/techlibrary/developing/bio-sensor/nivio-xmr-sensor.html). Compared to conventional MR sensors, this sensor offers the following novel features:
1.Ultra-high sensitivity and low noise characteristics enable highly accurate detection of weak magnetic signals.
2.The design is optimized for biomagnetic applications, making it possible to detect biomagnetic signals and magnetic nanoparticles.
3.Sensor array integration allows for simultaneous multi-point detection and high spatial resolution.
4.The wide dynamic range and excellent linearity enable quantitative measurements without the need for magnetic shielding.
By leveraging these features, this study successfully achieved magnetocardiography measurements outside of a magnetic shield, which was difficult with conventional technologies.
3. Work has no original graphical data except graphs. The photograph of the device and schematic description of the measurements involving patients would be an advantage.
Thank you for your suggestions. We added picture describing the measurements of MCG in Figure 1C.
4. Utilization of the magnetic field sensors of different types for detection and evaluation of caddio-vascular signals was previously widely discussed in the literature. Comparative general analysis and discussion around this problem might be important for this journal audience as different kind of magnetic sensors have both particular advantages and disadvantages starting from magnetic field sensitivity (Uchiyama, T.; Mohri, K.; Honkura, Y.; Panina, L.V. Recent advances of pico-Tesla resolution magneto-impedance sensor based on amorphous wire CMOS IC MI Sensor. IEEE Trans. Magn. 2012, 48, 3833–3839; Prinz, G.A. Magnetoelectronics applications. Magn. Magn. Mater.1999, 200, 57–68; Kozlov, N.V., Volchkov, S.O., Blyakhman, F.A., Chestukhin, V.V., Kurlyandskaya, G.V. The Modeling of Magnetic Detection of Iron Oxide Nanoparticles in the Stream of Patient-Specific Artery With Stenotic Lesion: The Effects of Vessel Geometry and Particle Concentration IEEE Transactions on Magnetics, 2022, 58(8), 5100305). There were excellent MCG measurements with portable SQUIDs like - Impedance magnetocardiography: Experiments and modeling by Claycomb. Discussion using this experience might be just mor convincing.
Thank you for giving your insight to our paper. Regarding the paper “Recent advances of pico-Tesla resolution magneto-impedance sensor based on amorphous wire CMOS IC MI Sensor”, we have mentioned the similar technology by induction coil in discussion. We added the paper by Uchiyama in this section for further discussion (Ref. 32). We added the paper " Magnetoelectronics applications " in materials and methods (Ref. 15). For the paper “The Modeling of Magnetic Detection of Iron Oxide Nanoparticles in the Stream of Patient-Specific Artery With Stenotic Lesion”, we recognized these paper focused on the detection of magnetic nanoparticles, which was different from our purpose. The paper “Impedance magnetocardiography: Experiments and modeling” utilized SQUID sensor with liquid nitrogen. We added these papers in Discussion (Ref. 26),
5. From the description of the patients, it becomes not very clear if they are supposedly healthy or having identified cardiac problems. In the last case the averaging should be done separately.
Thank you for pointing out. We excluded patients with cardiac disease in this study. We added the decription in Materials&Methods.
6. Figures 5 and 6 have clear technical deficiencies – they are not readable. Fig.5 part a) should be the double size; parts b) and c) maybe placed together in the next row. Fig.6 should be the double size of the present dimension.
We appreciate your comment. According to your comment, we replaced the figure with higher resolution.
7. Conclusions must contain numerical data.
Thank you for your valuable comments. We modified our conclusion with the signal-to-noise ratio of MCG.
Reviewer 3 Report
Comments and Suggestions for Authors
This manuscript develops a magnetocardiography system used magneto-resistive sensors without the magnetic shielded room. After reviewing the manuscript, following comments and questions has been raised to the authors:
- The intensity of geomagnetic field is 10-30 uT? As far as I know, the geomagnetic field is around 0.5 Gs in some place on the earth.
- The PSD curve (noise curve) should be given in section 2. The range of MR is -45-45 uT, is it enough for the geomagnetic field? Which direction of the sensitive axis of MR in the testing room, please give a schematic diagram for the convenience of reading.
- Line 112, the sampling rate is set as 5 kHz, but the frequency of MCG signal less than 100 Hz, why is the sampling rate so high?
- Line 121, please briefly explain the principle of ANC for easier reading.
- “No signal processing” in Figure 4 (a), but “after digital filter only (a)” in line 198, So what is it exactly? The quality of Figure 4 and 6 should be improved for reading.
Author Response
Reviewer 3
This manuscript develops a magnetocardiography system used magneto-resistive sensors without the magnetic shielded room. After reviewing the manuscript, following comments and questions has been raised to the authors:
1. The intensity of geomagnetic field is 10-30 uT? As far as I know, the geomagnetic field is around 0.5 Gs in some place on the earth.
Thank you for your valuable comment. As you pointed out, the intensity of the geomagnetic field varies depending on the location, but it is generally reported to be around 0.25–0.65 Gauss (25–65 μT). The value of "10–30 μT" described in the manuscript was inaccurate, and we will correct it to reflect the proper range. We corrected the description. We appreciate your helpful feedback.
2. The PSD curve (noise curve) should be given in section 2. The range of MR is -45-45 uT, is it enough for the geomagnetic field? Which direction of the sensitive axis of MR in the testing room, please give a schematic diagram for the convenience of reading.
The dynamic range of the sensor is specified as ±45 μT in the datasheet; however, the actual performance of the sensor allows for a range of approximately ±65 μT. Therefore, we consider the sensor to be sufficiently capable for measuring the geomagnetic field. The sensitive axis of the sensor is aligned with its longitudinal direction, which corresponds to the direction perpendicular to the backrest surface of the sofa in the testing environment.
3. Line 112, the sampling rate is set as 5 kHz, but the frequency of MCG signal less than 100 Hz, why is the sampling rate so high?
The reason for setting a high sampling rate is as follows. Although the frequency components of the MCG signal are below 100 Hz, a sampling rate of 5 kHz allows for more accurate capture of fine waveform details, including rapid rises and falls in the signal. In addition, acquiring data at a high sampling rate increases the flexibility of post-processing, such as the application of digital filters and noise reduction, and helps to prevent aliasing. Furthermore, the detection accuracy of event timing and peak positions is improved, leading to more reliable signal analysis. For these reasons, a sampling rate of 5 kHz was adopted in this study. We described them in Section 2.3
4. Line 121, please briefly explain the principle of ANC for easier reading.
The principle of adaptive noise cancelling (ANC) used in this study is as follows. A detection magnetic sensor acquires the primary signal, which contains both the desired signal and noise, while a reference magnetic sensor acquires a signal that primarily contains noise similar to that affecting the detection sensor. Using an adaptive algorithm, the noise component from the reference sensor is estimated and subtracted from the primary signal, thereby effectively reducing noise while preserving the desired signal. This approach enables more accurate extraction of the target signal from a noisy environment. We added the description in Section 2.4.
5. “No signal processing” in Figure 4 (a), but “after digital filter only (a)” in line 198, So what is it exactly? The quality of Figure 4 and 6 should be improved for reading.
Thank you for pointing out the error. We changed “no signal processing” to “digital filtere only” to unify the description.
Reviewer 4 Report
Comments and Suggestions for Authors
In the manuscript, the author developed a magnetocardiography (MCG) system based on a room-temperature magnetoresistive sensor, which uses a wide dynamic range magnetoresistive sensor and does not require magnetic shielding environments. Combining adaptive noise cancellation (ANC) and Bayesian signal space projection (SSP), the P-QRS-T waveform was successfully detected in the conventional environment, achieving an important breakthrough in unshielded MCG technology. However, there are still some problems to be solved before publication.
1、How does the system proposed by the author manage to reliably capture cardiac signals without magnetic shielding?
2、Line 112 on page 3: Why is the sampling rate set to 5000Hz? What is the basis for this?
3、Line 133 on page 4: The author did not clarify the details of each variable in the Bayesian formula, such as how YI (t) is extracted from environmental noise?
4、Line 165 on page 5: The majority of the experimental participants are male and their age distribution is concentrated. Will this affect the universality of the test results?
5、Line 191 on page 6: Is Bayesian SSP adapted to sudden noise interference?
6、Regarding cardiac signal processing, it is suggested that the author cite the literature "Deep Learning in Heart Sound Analysis: From Techniques to Clinical Applications".
Author Response
Reviewer 4
In the manuscript, the author developed a magnetocardiography (MCG) system based on a room-temperature magnetoresistive sensor, which uses a wide dynamic range magnetoresistive sensor and does not require magnetic shielding environments. Combining adaptive noise cancellation (ANC) and Bayesian signal space projection (SSP), the P-QRS-T waveform was successfully detected in the conventional environment, achieving an important breakthrough in unshielded MCG technology. However, there are still some problems to be solved before publication.
1、How does the system proposed by the author manage to reliably capture cardiac signals without magnetic shielding?
Thank you for raising an important point. In this method, external noise is canceled using the empty room noise obtained in advance. Therefore, although we have confirmed that it is possible to measure magnetic fields when there is a change in external noise such as a person walking around the examination room, it is impossible to extract magnetic field signals when sudden noise is introduced. We have added this point to the limitations.
2、Line 112 on page 3: Why is the sampling rate set to 5000Hz? What is the basis for this?
The reason for setting the sampling rate to 5000 Hz is as follows. Although the main frequency components of the MCG signal are below 100 Hz, a higher sampling rate allows for more accurate detection of external noise (such as power line interference, its harmonics, and high-frequency noise from equipment), which can then be effectively removed through digital filtering and adaptive noise cancellation. In addition, a higher sampling rate facilitates the design of anti-aliasing filters, ensures faithful signal reproduction, and improves the accuracy of post-processing analyses. For these reasons, a sampling rate of 5000 Hz was selected.
3、Line 133 on page 4: The author did not clarify the details of each variable in the Bayesian formula, such as how YI (t) is extracted from environmental noise?
Thank you for your valuable comment. We decided to add the phrase to show the details of Bayesian formula as follows:
where y(t) is a vector representing measured values of the MCG sensor array at time t. yS(t) is the signal vector containing the spatial information of the signal of interest. yI(t) represents the interference vector generated from sources outside the signal sources. ε denotes stationary noise that is uncorrelated across channels [19, 20]. Both yS(t) and yI(t) are components projected from factors, defined as yS(t) = A x(t), where A is the signal factor loading matrix and x(t) is the signal factor, and yI(t) = B u(t), where B is the interference factor loading matrix and u(t) is the interference factor. In PFA [cite-pfa], B is determined by applying the factor analysis method to ERN data. Once B is determined, A is estimated by applying the factor analysis method to MCG data, which includes both MCG and interference, by modeling the total factor loading matrix as [A; B], with B fixed to the value determined from the ERN analysis. With A and B determined, we can estimate x(t) and consequently the denoised signal component, yS(t).
4、Line 165 on page 5: The majority of the experimental participants are male and their age distribution is concentrated. Will this affect the universality of the test results?
When we performed a gender-specific SNR analysis on this dataset, we found that men tended to have a slightly higher SNR(men: 35.8 dB; women: 30.1dB), but the analysis could be performed without problems for women as well. In addition, no correlation was found between age and SNR. However, as the reviewer pointed out, the number of data was small and biased toward a certain demographic. We have noted this in the limitations. In our current investigation, we are including even older subjects, but we are able to record data without any problems, and we would like to clarify this in the future.
5、Line 191 on page 6: Is Bayesian SSP adapted to sudden noise interference?
Thank you for your comments. Since Bayesian SSP assumes that the spatial arrangement of noise sources does not change over time, it is difficult to cancel sudden noise interference. We added this sentence in Limitation section.
6、Regarding cardiac signal processing, it is suggested that the author cite the literature "Deep Learning in Heart Sound Analysis: From Techniques to Clinical Applications".
Thank you for the suggestion. Although we did not use it in this study, application of deep learning would improve the data analysis, we cited it in discussion (Ref. 44).
Reviewer 5 Report
Comments and Suggestions for Authors
In the manuscript "Recording of Cardiac Excitation using a Novel Magnetocardiography System with Magnetoresistive Sensors outside a Magnetic Shielded Room" the authors proposed a 6*7 magnetoresistive sensor array for magnetic cardiography (MCG). Each sensor is 4 magnetoresistive elements were assembled to form a Wheatstone bridge to improve the signal-to-noise ratio. In addition, various post-processing of the signal was used to reduce noise. The authors tested the invention on 40 patients. In general, good agreement with electrocardiography data was obtained. The manuscript, except for several shortcomings, is well written and will be of interest to readers of Sensors.
However, before recommending a manuscript for publication, the Reviewer invites authors to pay attention to the following comments.
- The Reviewer believes that it would be useful to discuss MKG based on magnetoimpedance sensors in the Introduction. Professor T. Uchiyama's group has made great progress in this area (see, for example, 10.1063/1.4975128 or 10.1016/j.jmmm.2020.167148).
- The authors need to expand the Conclusion. It should reflect the results obtained.
- There are typos in the manuscript. For example, lines 69-70: “We developed novel magnetic resonance sensors…”. Apparently, the authors mean “magnetoresistive sensors”.
- Lines 90-91. The authors write "The electrical resistance of the sensor element varies depending on the angle between the sensor element and the external magnetic field…". The reviewer believes that this phrase does not reflect the essence of the phenomenon. It would be more accurate to say that the electrical resistance of the sensor element varies depending on the external magnetic field value. The dependence on the angle is due to the change in the projection of the field onto the sensor axis.
- The reviewer recommends that the authors describe in more detail how the ECG was performed (equipment, procedure, etc.).
- The patients varied considerably in height and weight. So, the question arises, were their hearts oriented the same relative to the sensor? Were the distances from the hearts to the sensor the same? Could this be related to the rather large scatter of data in Table 2? It is worth discussing this in the manuscript.
- What is the reason for the noticeable difference between P- and T amplitudes on MCG and ECG?
- Authors need to pay attention to the Figures. For example, Fig. 3b is of very poor quality, the inscriptions are very small.
Author Response
Reviewer 5
In the manuscript "Recording of Cardiac Excitation using a Novel Magnetocardiography System with Magnetoresistive Sensors outside a Magnetic Shielded Room" the authors proposed a 6*7 magnetoresistive sensor array for magnetic cardiography (MCG). Each sensor is 4 magnetoresistive elements were assembled to form a Wheatstone bridge to improve the signal-to-noise ratio. In addition, various post-processing of the signal was used to reduce noise. The authors tested the invention on 40 patients. In general, good agreement with electrocardiography data was obtained. The manuscript, except for several shortcomings, is well written and will be of interest to readers of Sensors.
However, before recommending a manuscript for publication, the Reviewer invites authors to pay attention to the following comments.
1. The Reviewer believes that it would be useful to discuss MKG based on magnetoimpedance sensors in the Introduction. Professor T. Uchiyama's group has made great progress in this area (see, for example, 10.1063/1.4975128 or 10.1016/j.jmmm.2020.167148).
Thank you for giving your insight to our paper. Regarding the paper “Recent advances of pico-Tesla resolution magneto-impedance sensor based on amorphous wire CMOS IC MI Sensor”, we have mentioned the similar technology by induction coil in discussion. We added the paper by Uchiyama in this section for further discussion (Ref. 32).
2. The authors need to expand the Conclusion. It should reflect the results obtained.
Thank you for valuable comments. We modified our conclusion with the signal-to-noise ratio of MCG.
3. There are typos in the manuscript. For example, lines 69-70: “We developed novel magnetic resonance sensors…”. Apparently, the authors mean “magnetoresistive sensors”.
We apologize the error. We corrected it.
4. Lines 90-91. The authors write "The electrical resistance of the sensor element varies depending on the angle between the sensor element and the external magnetic field…". The reviewer believes that this phrase does not reflect the essence of the phenomenon. It would be more accurate to say that the electrical resistance of the sensor element varies depending on the external magnetic field value. The dependence on the angle is due to the change in the projection of the field onto the sensor axis.
We appreciate your valuable comments. We rewrote the phrase according to your comments: “The electrical resistance of the sensor element varies depending on the angle between the sensor axis and the projection of the external magnetic fields”.
5. The reviewer recommends that the authors describe in more detail how the ECG was performed (equipment, procedure, etc.).
Non-magnetic electrodes (V-09IO3, NIHON KODEN) with conductive adhesive gel and carbon cables were used for ECG measurements. The ECG circuit was a custom-built two-lead electrocardiograph, designed with reference to Texas Instruments ICs and application notes (https://www.ti.com/solution/en-jp/electrocardiogram-ecg#tech-docs). Analog signals were digitized using an NI-9202 A/D converter (National Instruments). Subjects were recorded in a resting state, and the electrodes were placed in a standard two-lead configuration. The ECG was recorded simultaneously with the MCG measurement for two minutes. The obtained ECG data were used for comparison with the MCG data. We added this description in Method section.
6. The patients varied considerably in height and weight. So, the question arises, were their hearts oriented the same relative to the sensor? Were the distances from the hearts to the sensor the same? Could this be related to the rather large scatter of data in Table 2? It is worth discussing this in the manuscript.
Thank you for raising the important issue. The STORM system does not have a mechanism for adjusting the position of the sensor array. Thus as the reviewer pointed out, the orientation of the heart varied from subject to subject. We mentioned this issue in the Limitation section. According to the reviewer’s comment, we added the sentence "The quality of the results could be enhanced by further improvement of the seat that allows for positional adjustments of the sensor array or the participant during MCG recording."
7. What is the reason for the noticeable difference between P- and T amplitudes on MCG and ECG?
In this study, the amplitudes of the P and T waves in the magnetocardiogram (MCG) were lower than those in the electrocardiogram (ECG). This discrepancy can be attributed to two main factors. First, the ECG recordings were obtained using lead II, a lead commonly used for its clear visualization of P and T waves, whereas the MCG did not necessarily utilize a channel optimized for the visibility of these waves. Second, in the MCG analysis, the channel was selected based on the maximum QRS amplitude, without specific consideration for the clarity or magnitude of the P and T waves. We added new paragraph to discuss in Discussion.
8. Authors need to pay attention to the Figures. For example, Fig. 3b is of very poor quality, the inscriptions are very small.
Thank you for valuable comment. We changed the figure with higher resolution.
Round 2
Reviewer 2 Report
Comments and Suggestions for Authors
Work was sufficiently improved and it can be published in the present state.
Reviewer 3 Report
Comments and Suggestions for Authors
The author have answered all of my questions. I think the journal can accept this manuscript.
Reviewer 5 Report
Comments and Suggestions for Authors
The authors took into account the Reviewer's comments and significantly revised the manuscript. The manuscript can be recommended for publication.